# GraphCroc: Cross-Correlation Autoencoder for Graph Structural Reconstruction

**Shijin Duan**[*] **Ruyi Ding**[*] **Jiaxing He** **Aidong Adam Ding** **Yunsi Fei** **Xiaolin Xu**
Northeastern University
{duan.s, ding.ruy, he.jiaxi, a.ding, y.fei, x.xu}@northeastern.edu

## Abstract

Graph-structured data is integral to many applications, prompting the development of various graph representation methods. Graph autoencoders (GAEs), in particular, reconstruct graph structures from node embeddings. Current GAE models primarily utilize self-correlation to represent graph structures and focus on node-level tasks, often overlooking multi-graph scenarios. Our theoretical analysis indicates that self-correlation generally falls short in accurately representing specific graph features such as islands, symmetrical structures, and directional edges, particularly in smaller or multiple graph contexts. To address these limitations, we introduce a cross-correlation mechanism that significantly enhances the GAE representational capabilities. Additionally, we propose the GraphCroc, a new GAE that supports flexible encoder architectures tailored for various downstream tasks and ensures robust structural reconstruction, through a mirrored encoding-decoding process. This model also tackles the challenge of representation bias during optimization by implementing a loss-balancing strategy. Both theoretical analysis and numerical evaluations demonstrate that our methodology significantly outperforms existing self-correlation-based GAEs in graph structure reconstruction. Our code is available in https://github.com/sjduan/GraphCroc.

## 1 Introduction

Graph-structured data captures the relationships between data points, effectively mirroring the inter-connectivity observed in various real-world applications, such as web services [3], recommendation systems [39], and molecular structures [17, 18, 12]. Beyond the message passing through node connections [50], the exploration of graph structure representation is equally critical [38, 15, 33, 19, 9]. This representation is extensively utilized in domains including recommendation systems, social network analysis, and drug discovery [42], by leveraging the power of Graph Neural Networks (GNNs). Specifically with $L$ layers in GNN, a node assimilates structural information from its $L$-hop neighborhood, embedding graph structure in node features.

Graph autoencoders (GAEs)[19] have been developed to encode graph structures into node embeddings and decode these embeddings back into structural information, such as the adjacency matrix. This structural reconstruction process can be performed either sequentially along nodes [11, 22, 47] or in a global fashion [33]. While there has been significant advancement in both methods, most studies primarily focus on node tasks, which involve a single graph, such as link prediction [35] and node classification [15], with decoding strategies typically reliant on "self-correlation". We define this term as the correlation of node pair from the same node embedding space. Given an $n$-node graph with embedding dimension $d'$ on each node, its node embedding is $Z \in \mathbb{R}^{n \times d'}$, thus the self-correlation is expressed as $z_i^T z_j$ between two nodes. Correspondingly, the "cross-correlation" depicts the node pair

---

[*]equal contribution

38th Conference on Neural Information Processing Systems (NeurIPS 2024).

correlation as $p_i^T q_j$, which are from two separate embedding spaces, $P, Q \in \mathbb{R}^{n \times d'}$. However, studies are seldom evaluated under graph tasks, which represent graph structure on multiple graphs. The distinctiveness of each graph presents a significant challenge in accurately representing all graphs.

In this work, we demonstrate the limitations of self-correlation in structure representation, such as accurately representing islands, topologically symmetric graphs, and directed graphs. Although these deficiencies may appear infrequently in large, undirected single graphs, they are prevalent and critical in smaller to moderately-sized multiple graphs, e.g., molecules [12]. Conversely, we establish that decoding based on cross-correlation can significantly mitigate these limitations, offering improvements in both undirected and directed graphs. Furthermore, the optimization of self-correlation-based GAE has to proceed in a restricted space. On the other hand, the cross-correlation can double the argument space that is not restricted during optimization. It makes the region of attraction smoother and easier to converge, indicating the superior representational ability of cross-correlation.

Accordingly, we propose a novel GAE model, namely GraphCroc, which leverages cross-correlation to node embeddings and a U-Net-like encoding-decoding procedure. Previous GAE models carefully design the encoder for a faithful structure representation, yet keep the decoder as the straightforward node correlation computation. Differently, GraphCroc retains the freedom of encoder design, facilitating its architecture design for downstream tasks, rather than structure representation. We define the decoder as a mirrored architecture of the encoder, to gradually reconstruct the graph structure. The encoder shapes the down-sampling of GraphCroc, while the decoder half performs the up-sampling for structural reconstruction. In addition, regarding the unbalanced population of zeros and ones in graph structure, i.e., sparse adjacency matrix, we define loss balancing on node connections.

We highlight our contributions as follows:

- We analyze the representational capabilities of self-correlation within GAE encoding, highlighting its limitations. Furthermore, we elaborate how cross-correlation addresses these deficiencies, facilitating a smoother optimization process.

- We propose GraphCroc, a cross-correlation-based GAE that integrates seamlessly with GNN architectures for graph tasks as its encoder, and structures a mirrored decoder. GraphCroc offers superior representational capabilities, especially for multiple graphs.

- To the best of our knowledge, this is the first evaluation of structural reconstruction using GAE models on graph tasks. Besides, we assess the performance of our GraphCroc model integrated with other GAE strategies and on various downstream tasks.

- We evaluate the potential of GraphCroc in domain-specific applications, such as whether a GAE focused on structural reconstruction could be an attack surface for edge poisoning attacks, given the effectiveness and stealth of adversarial attacks using AE in vision tasks.

## 2 GAE Structural Reconstruction Analysis

### 2.1 Preliminary

**Graph Neural Network**   As Graph Neural Networks (GNNs) have been defined in various ways, without loss of generality, we adopt the computing flow in [45] to define the graph structure and a general GNN model. A graph $G = (V, E)$ comprises $n$ nodes, each with a feature represented by a $d$-dimensional vector, resulting in a feature matrix $X \in \mathbb{R}^{n \times d}$. The set of edges $E$ is depicted by an adjacency matrix $A \in \{0, 1\}^{n \times n}$, which indicates the connections between nodes in $G$. A GNN model $f(X, A)$ is utilized to summarize graph information for downstream tasks.

The feed-forward propagation of the $l$-th layer in GNN $f(\cdot)$ is

$$h_{l+1} = \sigma \left( \hat{D}_l^{-\frac{1}{2}} \hat{A}_l \hat{D}_l^{-\frac{1}{2}} h_l W_l \right) \tag{1}$$

$h_l \in \mathbb{R}^{n \times d_l}$ is the input feature of the $l$-th layer, where $h_1 = X$ and $d_l$ is the feature dimension of each node specified by each layer. $\hat{A}_l = A_l + I$ is the adjacency matrix (self-loop added) of the input graph structure in each layer. Note that $\hat{A}_l$ will be consistent in the absented pooling layer scenario, such as the node classification task, yet it can vary along GNN layers in graph tasks due to the introduction of graph pooling layers [9]. Subsequently, we use the diagonal node degree matrix

$\hat{D}_l$ of $\hat{A}_l$ to normalize the aggregation. For layer components, $W_l$ is the weight matrix and $\sigma$ is the activation function in the $l$-th layer.

**Graph Autoencoder** The naïve GAE is proposed to reconstruct the adjacency matrix $A$ of a graph $G$ through node embedding $Z$:

$$\text{encoder: } Z = \Phi(Z|G) = f(X, A), \quad \text{decoder: } \tilde{A} = \Theta(A|Z) = \text{sigmoid}(ZZ^T) \tag{2}$$

GAE first encodes the graph $G$ through a general GNN model $f(\cdot)$ (a.k.a. inference model), resulting in node embeddings in the latent space $Z \in \mathbb{R}^{n \times d'}$ where $d'$ is the latent vector dimension for each node. The generative model is non-parameterized, but the tensor product of the node embeddings. It measures the correlation between each node pairs, i.e., $\text{sigmoid}(z_i^T z_j)$, normalized as the link probability by the logistic sigmoid function. Usually, the predicted connection can be defined as an indicator function $\mathcal{I}(\tilde{A}_{i,j}) = \mathbb{I}(\tilde{A}_{i,j} \geq th)$, e.g., $th = 0.5$.

With the same functionality, GAE has been improved with other enhancements, such as variational embedding [19], MLP-based decoder [33], masking on features [15] and edges [23]. Note that other correlation measurements in the decoder are also proposed, such as Euclidean distance between node embedding [26], i.e., $\tilde{A}_{i,j} = \text{sigmoid}(C(1 - \|z_i - z_j\|_2^2))$, where $C$ is a temperature hyperparameter. In general, they predict the connection between node pairs by measuring the similarity, such as inner product and L2-norm, from the same embedding space. Related work is discussed in Appendix A.

## 2.2 Deficiencies of Self-Correlation on Graph Structure Representation

To generalize the discussion, we set the representation capability of the encoder as unrestricted, i.e., allowing $Z$ to be generated through any potentially optimal encoding method. On the decoding side, self-correlation is applied following Eq.2. We identify and discuss specific (sub)graph structures that are poorly addressed by current self-correlation methods:

**Islands (Non Self-Loop).** Both the inner product and the L2-norm of a node embedding pair fail to accurately represent nodes without self-loops, where $A_{i,i} = 0$. Given that $z_i^T z_i \geq 0$ and $C(1 - \|z_i - z_i\|_2^2) = C > 0$, the predicted value $\mathcal{I}(\tilde{A}_{i,i})$ defaults to 1 when threshold 0.5 is applied to the sigmoid output. This limitation underlies the common homophily assumption in previous research [42], that all nodes contain self-loops; it treats self-loops as irrelevant to the graph's structure. However, there is a huge difference between the self-connection on one node and the inter-connection between nodes in some scenarios; for example, on heterophilous graphs [51], nodes are prone to connect with other nodes that are dissimilar — such as fraudster detection.

**Topologically Symmetric Structures.** If graph structure is symmetric along an axis or a central pivot, as demonstrated in Figure 1, the self-correlation method cannot represent these structures as well.

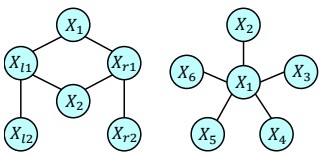

**Definition 2.1** (Topologically Symmetric Graph). A symmetric graph $G = (V, E)$ has the structure and node features topologically symmetric either along a specific axis or around a central node. For an axisymmetric graph, the feature matrix is denoted as $X = \{X_1, \ldots, X_{n_1}\} \cup \{X_{l1}, \ldots, X_{ln_2}\} \cup \{X_{r1}, \ldots, X_{rn_2}\}$, such that $n_1 + 2n_2 = n$. $X_i$ represents the node features on the axis, and for each paired node off the axis, $X_{li} = X_{ri}$. The connections are also symmetric, satisfying $A_{li,:} = A_{ri,:}$. In a centrosymmetric graph, the

Figure 1: Two examples of the topological symmetric graphs. The left graph is axissymmetric; the right graph is centrosymmetric.

pivot node has feature $X_1$, and other nodes share the same feature $X_2 = \ldots = X_{n-1}$. Additionally, the adjacency relationships for these nodes are identical, with $A_{i,:} = A_{j,:}$ for all $i, j \in [2, n]$.

**Lemma 2.2.** *Given an arbitrary topologically symmetric graph $G = (V, E)$ and an encoder $f(X, A)$, the self-correlation decoder output will always have $\mathcal{I}(\tilde{A}_{li,ri}) = 1$.*

*Proof.* Due to the symmetry on graph $G = (V, E)$ from the definition above, we have $f(X_i, A_{i,:}) = f(X_j, A_{j,:})$ for nodes $i$ and $j$ that are symmetric about the axis or pivot. Thus, we can derive $z_i = z_j$. For the prediction on link between $i$ and $j$, we have $\tilde{A}_{i,j} = \text{sigmoid}(z_i^T z_j) = \text{sigmoid}(z_i^T z_i) \geq 0.5$. Similarly, for the L2-norm method, we have $\tilde{A}_{i,j} = \text{sigmoid}(C(1 - \|z_i - z_j\|_2^2)) = \text{sigmoid}(C(1 - \|z_i - z_i\|_2^2)) = \text{sigmoid}(C) > 0.5$. In both decoding methods, the decoder is prone to predict the edge between two symmetric as positive, $\mathcal{I}(\tilde{A}_{i,j}) = 1$. $\square$

Consequently, the self-correlation method will indicate a positive connection between two symmetric nodes, regardless of their real connection status on the graph $G$. Note that for variational GAE, $Z$ is sampled from a Gaussian distribution whose mean and variance come from GNN models; thus it still follows the above lemma, even if randomness is involved during the encoding.

**Directed Graph.** Another straightforward deficiency of the self-correlation method is that it always generates a symmetric adjacency matrix, $\tilde{A}$, which is not suitable for directed graphs that require an asymmetric adjacency matrix. This issue is also acknowledged in [20], which proposes a solution involving dual encoding using the Weisfeiler-Leman (WL) algorithm [41] and node label coloring. However, this solution constrains the encoder to a dual-channel structure while maintaining the same decoding method as described in Eq. 2. This restriction can limit the flexibility of the encoder architecture, making it less adaptable for various downstream tasks.

Furthermore, we conduct a theoretical analysis of the dimensional requirements of node embedding to represent graph structures using self-correlation, in Appendix B.

### 2.3 Our Cross-Correlation Approach for Better Structural Representation

Instead of self-correlation, we advocate cross-correlation to reconstruct the graph structure, denoted by $\tilde{A} = \text{sigmoid}(PQ^T)$, where $P, Q \in \mathbb{R}^{n \times d'}$. This approach allows us to decouple the variables involved in calculating the correlation, thus overcoming the inherent limitations of self-correlation.

#### 2.3.1 How Does Cross-Correlation Mitigate Deficiencies of Self-Correlation?

**Expressing Islands.** For each node $i \in [1, n]$, the sign of $p_i^T q_i$ can be flexibly determined by $p_i$ and $q_i$, allowing it to be either positive or negative. Consequently, the presence of an island can be effectively modeled using $\tilde{A}_{i,j} = \text{sigmoid}(p_i^T q_i)$ or $\text{sigmoid}(C(1 - \|p_i - q_j\|_2^2))$. This approach avoids the limitations associated with self-correlation, which restricts the sigmoid input to positive.

**Expressing Symmetric Structure.** Cross-correlation is particularly effective in capturing topological symmetric structures. Given a node pair $(i, j)$ that is topologically symmetric about an axis or pivot, for undirected graphs, we have $p_i = p_j$ and $q_i = q_j$. However, since $p_i$ and $q_j$ (as well as $p_j$ and $q_i$) are not directly dependent on each other, the sign of $p_i^T q_j = p_j^T q_i$ can be either positive or negative. Therefore, $\mathcal{I}(\tilde{A}_{i,j}) = \mathcal{I}(\text{sigmoid}(p_i^T q_j))$ is able to yield 0 or 1, depending on the specific values of node embedding $p_i$ and $q_j$. This flexibility can be supported by the L2-norm decoding as well.

**Expressing Directed Graph.** A similar interpretation extends to the representation of directed graphs. For two nodes $i$ and $j$, the directed edges can be defined by $p_i^T q_j$ for one direction and $p_j^T q_i$ for the other. Since these four latent vectors do not have explicit dependencies among them, the directions of the edges can be independently determined using cross-correlation, capturing the directional connections between nodes.

In Appendix C, we further discuss and visualize how our cross-correlation approach represents these specific graph structures.

#### 2.3.2 Cross-Correlation Provides Smoother Optimization Trace

We highlight the superiority of cross-correlation over self-correlation in the optimization process of GAE training. Considering the decoder optimization problem, where we aim to satisfy the constraints:

$$\textbf{(self-correlation) } \mathcal{I}(\text{sigmoid}(z_i^T z_j)) = A_{i,j}, \textbf{ (cross-correlation) } \mathcal{I}(\text{sigmoid}(p_i^T q_j)) = A_{i,j} \quad (3)$$

for each element in matrix $A$. This involves finding $Z \in \mathbb{R}^{n \times d'}$ or $P, Q \in \mathbb{R}^{n \times d'}$ that maximize the number of satisfied constraints. Additionally, for an undirected graph, the symmetry $A_{i,j} = A_{j,i}$ imposes the requirement that $\mathcal{I}(\text{sigmoid}(p_i^T q_j)) = \mathcal{I}(\text{sigmoid}(p_j^T q_i))$, ensuring that both $p_i^T q_j$ and $p_i^T q_j$ should have the same sign.

In the case of cross-correlation, where $P$ and $Q$ are independently determined, and all constraints can well align with the generation of $P$ and $Q$. However, this is not the case in self-correlation, where $z_i^T z_j = z_j^T z_i$ inherently overloads the symmetry constraint. For example, if $A_{i,j} = A_{j,i} = 1$, we only require $p_i^T q_j > 0$ and $p_j^T q_i > 0$ for cross-correlation, while they become restrictive as

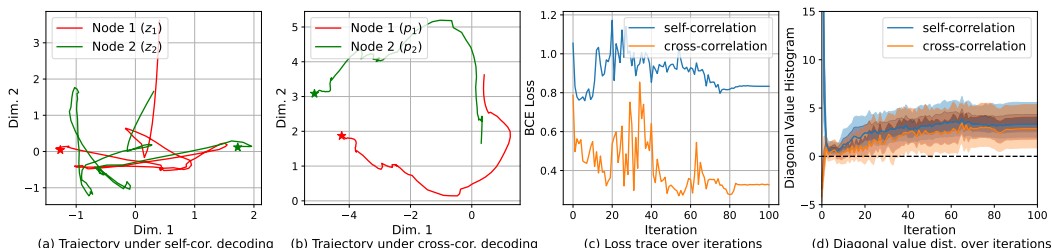

Figure 2: Training comparison between self-correlation and cross-correlation on PROTEINS subset (64 graphs). In (a) and (b), we demonstrate the trajectory of the first two node embeddings in the first graph during training iteration, where the star mark is the end-point of training. We apply PCA for dimension compression and the Savitzky-Golay filter to help trace visualization. We also set $z_i = p_i \neq q_i$ at the beginning of optimization to ensure that the traces of $z_i$ in (a) and $p_i$ in (b) start from the same point. (c) provides the BCE loss trace of this graph during training, showing that cross-correlation can lead the reconstruction to a better solution. (d) demonstrates the distribution of diagonal elements during training, i.e., $z_i^T z_i$ for self-correlation and $p_i^T q_i$ for cross-correlation. The results of other graphs are provided in Appendix G.2.

$z_i^T z_j = z_j^T z_i > 0$ in self-correlation. Cross-correlation offers a broader argument space, providing more freedom to find solutions that satisfy the constraints for reconstructing $A$. By employing gradient method during optimization, this process can be understood as the trajectory of $P$ and $Q$ being unrestricted in the argument space $\mathbb{R}^{n \times 2d'}$, while the trajectory of $Z$ is confined within a restricted space as $\mathbb{R}^{n \times d'}$. Therefore, cross-correlation facilitates smoother and more efficient convergence during optimization. Note that this restriction comes from the nature of self-correlation and cross-correlation, which is interpreted as the space reduction. During optimization, we can first find (local) optimal $P$ and $Q$, then apply $(PQ^T + QP^T)/2$ to interpret it as a symmetric matrix $\tilde{A}$, like $ZZ^T$; this procedure can still outperform direct optimization on $ZZ^T$.

**Validation on PROTEINS**   Here, we make a quick validation of our discussion on a subset of PROTEINS [2] that is a graph task, in Figure 2. By comparing Figure 2(a) and (b), the trajectory of node embedding during training demonstrates the superiority of cross-correlation over self-correlation. As the node embeddings are evolving under self-correlation, they frequently change their direction; this indicates that the region of attraction between the start and end point is not smooth and may follow a restricted manifold, thus the embedding cannot well converge to the local minimum. On the other hand, the region of attraction under cross-correlation is much smoother and easy to guide the node embedding to a better solution. This can also be evidenced by the lower loss under cross-correlation in Figure 2(c). Another concern of cross-correlation is that $p_i^T q_i$ cannot naturally satisfy $A_{i,i} = 1$ while $z_i^T z_i$ in self-correlation is able to. Nevertheless, $p_i^T q_i$ can be encouraged to perform with positive sign, proved by Figure 2(d), that all the diagonal element reconstruction $p_i^T q_i$ under cross-correlation can achieve positive sign at the end of training, leading to $\mathcal{I}(\text{sigmoid}(p_i^T q_i)) = A_{i,i} = 1$.

## 3   GraphCroc: Cross-Correlation-Based Graph Autoencoder

**Encoder Architecture**   Our work scales the normal single-graph representation to multi-graph for graph tasks, and we do not specify the encoder structure, but free it as the downstream tasks require. For graph tasks, the GNN model has a sequence of message passing layer and pooling layer to coarsen the graph to higher-level representation. In the end, a readout layer [8, 9, 31] is applied to summarize the graph representation to the latent space. We define the encoder as

$$\text{encoder (ours): } Z' = \Phi(Z'|G) = f(X, A) \qquad (4)$$

where $Z' \in \mathbb{R}^{n' \times d'}$ has a reduced number $n'$ of node embeddings. Besides, we exclude the readout layer from the encoder, yet assign it to the start of downstream tasks.

**Two-Way Decoder for Cross-Correlation**   To separately produce two node embeddings, $P$ and $Q$, we divide the decoding process into two parallel and self-governed decoders. Unlike [20], we leave

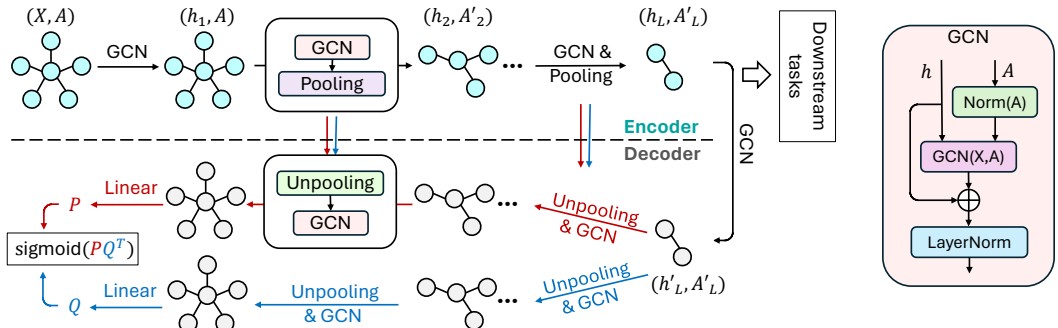

Figure 3: GraphCroc architecture. The encoder is generally demonstrated as a $L + 1$-layer GNN. The decoder has two paths to generate the node embedding for cross-correlation; each decoder is a mirrored structure of the encoder. Each decoder layer accepts the node feature and graph structure information from the corresponding encoder layer. Notably, the GCN module shown on the right incorporates skip connections and normalization to improve performance.

the encoder design to better suit specific downstream tasks, and focus primarily on the reconstruction challenges within the decoder design. Still with the latent vector $Z = f(X, A)$ generated by the encoder, we define our decoder as follow:

$$\textbf{decoder (ours):} \ \tilde{A} = \text{sigmoid}(PQ^T), \quad P = g_1(Z, \{A', h'\}), \quad Q = g_2(Z, \{A', h'\}) \quad (5)$$

$g_1(\cdot)$ and $g_2(\cdot)$ are two individual GNNs with the same structure, which take as input the latent vectors $Z$, the adjacency matrix groups $\{A'\}$ and node feature groups $\{h'\}$ from the encoder (as discussed in Section 3). Here, $\{A', h'\}$ is required for graph tasks that involves pooling/unpooling. In Appendix D, we further discuss why the two embeddings, $P$ and $Q$, in our decoder do not converge to each other, thereby preventing convergence to self-correlation decoding.

**Autoencoder Architecture**  In Figure 3, we present the autoencoder architecture, GraphCroc. The design of the encoder/decoder pair is inspired by the Graph U-Net structure [9], originally proposed for classification tasks. While the cross-correlation kernel remains unchanged, the encoder/decoder configuration can be tailored to various applications. Nevertheless, we utilize the Graph U-Net structure as a case study to demonstrate the effectiveness of cross-correlation.

The GraphCroc architecture follows the encoder formulations from Eq. 4 for multiple graphs and the decoder from Eq.5. During the encoding process, the graph architecture $A'$ is captured at each layer, which is then utilized in the corresponding unpooling layers to reconstruct the graph structure, as detailed in [9]. Additionally, skip connections enhance the model by capturing the node features $h'$ at each encoder layer and integrating them — either through addition or concatenation — into the node features of the corresponding decoder layer. Importantly, we emphasize the significance of implementing layer normalization [1] following each GNN layer. Although often overlooked in previous GAE studies due to the typically small number of layers, layer normalization is crucial for regulating gradient propagation as the number of layers increases.

**Loss Balancing**  The training on GAE is highly biased given the sparsity of the adjacency matrix. In practice, it is quite common that zero elements in $A$ take the majority, e.g., around $90\%$ in PROTEINS [2]. For a certain $A$ and its estimation $\tilde{A}$, we denote the vanilla loss function as $\mathcal{L} = \sum_{i=1}^{c_0} \mathcal{L}_i^0 + \sum_{j=1}^{c_1} \mathcal{L}_j^1$, where in total $c_0$ zeros and $c_1$ ones in $A$, and $\mathcal{L}^0/\mathcal{L}^1$ is their corresponding loss on each element. Since $c_0 \gg c_1$, the zero part dominates the loss function, inducing the decoder to predict zeros. Thus, we apply a scaling factor for each item:

$$\mathcal{L}(\tilde{A}, A) = \alpha_0 \sum_{i=1}^{c_0} \mathcal{L}_i^0 + \alpha_1 \sum_{j=1}^{c_1} \mathcal{L}_j^1, \quad \alpha_0 = \frac{c_0 + c_1}{2c_0}, \quad \alpha_1 = \frac{c_0 + c_1}{2c_1} \quad (6)$$

The derivation is provided in Appendix E.

Table 1: The AUC score of reconstructing the adjacency matrix in graph tasks. We reproduce the most representative global GAE methods with different decoding strategies. The self-correlation methods include naïve GAE, variational GAE [19], L2-norm (EGNN) [30], and our GraphCroc under self-correlation; the cross-correlation methods include directed representation (DiGAE) [20] and our GraphCroc. The best results are in **bold**, and the second bests are underlined.

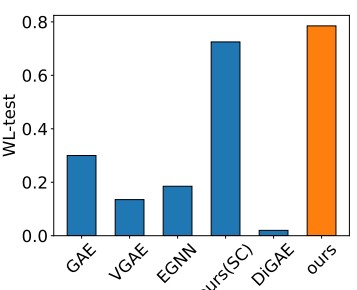

Figure 4: WL-test results on different GAE methods, in the IMDB-B task.

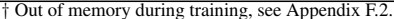

|  | Self-Correlation | | | | Cross-Correlation | |
|---|---|---|---|---|---|---|
|  | GAE | VGAE | EGNN | **GraphCroc**(SC) | DiGAE | **GraphCroc** |
| PROTEINS | 0.4750 | 0.4764 | 0.9608 | 0.9781 | 0.7577 | **0.9958** |
| IMDB-B | 0.7556 | 0.7105 | 0.9873 | 0.9892 | 0.7500 | **0.9992** |
| Collab | 0.7885 | 0.7946 | 0.9947 | 0.9926 | 0.7973 | **0.9989** |
| PPI | 0.6330 | 0.6239 | $-^{\dagger}$ | 0.9764 | 0.8364 | **0.9831** |
| QM9 | 0.5376 | 0.4852 | 0.9984 | 0.9967 | 0.7791 | **0.9987** |

† Out of memory during training, see Appendix F.2.

## 4 Evaluation

### 4.1 Experimental Setup

**Dataset** We assess GraphCroc in various graph tasks. Specifically, we utilize datasets for molecule, scaling from small (PROTEINS [2]) to large (Protein-Protein Interactions (PPI) [12], and QM9 [29]), for scientific collaboration (COLLAB [46]), and for movie collaboration (IMDB-Binary [46]). Further details on these datasets are provided in Appendix F.1.

**GraphCroc Structure** During evaluation, our GraphCroc structure, especially the GCN layer number, is determined by the scale of graph. Assuming the average node number in a graph task is $\bar{n}$, we set up an empirical layer number $L$ as $\left\lfloor \bar{n} \cdot \Pi_{i=1}^{L}(0.9 - i) \right\rfloor = 2$, so that the number of nodes can be smoothly reduced to two at the end of encoding [9]. Besides, we use node embedding dimension $d' \approx \bar{n}$, following analysis in Appendix B. Other detail is provided in Appendix F.2.

### 4.2 GraphCroc on Structural Reconstruction

Table 1 demonstrates the graph reconstruction capabilities of GAE models for multi-graph tasks by presenting ROC-AUC scores on adjacency matrix reconstruction. We compare our model against prevalent self-correlation kernels in GAEs and a cross-correlation method designed for directed graphs. Comparing the basic inner-product methods, GAE and VGAE [19], the variational extension in VGAE does not obviously improve the performance over GAE in graph tasks. However, by enhancing the GAE architecture itself, i.e., using our GraphCroc architecture, the reconstruction efficacy is significantly increased; GraphCroc under self-correlation can achieve 3/5 second bests among all GAE models. This underscores the graph representation capability of the U-Net structure [9]. Additionally, the L2-norm decoding method generally outperforms the inner-product approach (GAE and VGAE), although it struggles with large graphs such as PPI, which requires too much GPU memory to be trained during our reproduction. On the other hand, the cross-correlation method (DiGAE) provides a consistent albeit modest representation across different graph sizes. This demonstrates the cross-correlation ability to represent multiple graphs in various scenarios. However, the GNN architecture limits its capability to capture enough structural information. By integrating the strengths of cross-correlation with the U-Net architecture, our GraphCroc GAE model consistently excels over other methods, offering significant advantages in all the graph tasks tested. Even on large graphs, such as PPI with over 2,000 nodes, GraphCroc can still achieve an AUC score over 0.98. To further demonstrate the effectiveness of the cross-correlation mechanism, we evaluate GAE models with alternative architectures under cross-correlation, in Appendix G.1. The reconstruction results for these architectures are consistent with those in Table 1, though they exhibit slightly lower AUC compared to GraphCroc.

While the AUC score indicates how well a model reconstructs edges, it does not measure the model's ability to exactly reconstruct a whole graph. To address this, we employ the Weisfeiler-Leman isomorphism test (WL-test)[41], which assesses the structural equivalence between the original and reconstructed graphs. Figure 4 presents normalized WL-test results, i.e., the pass rates across all test

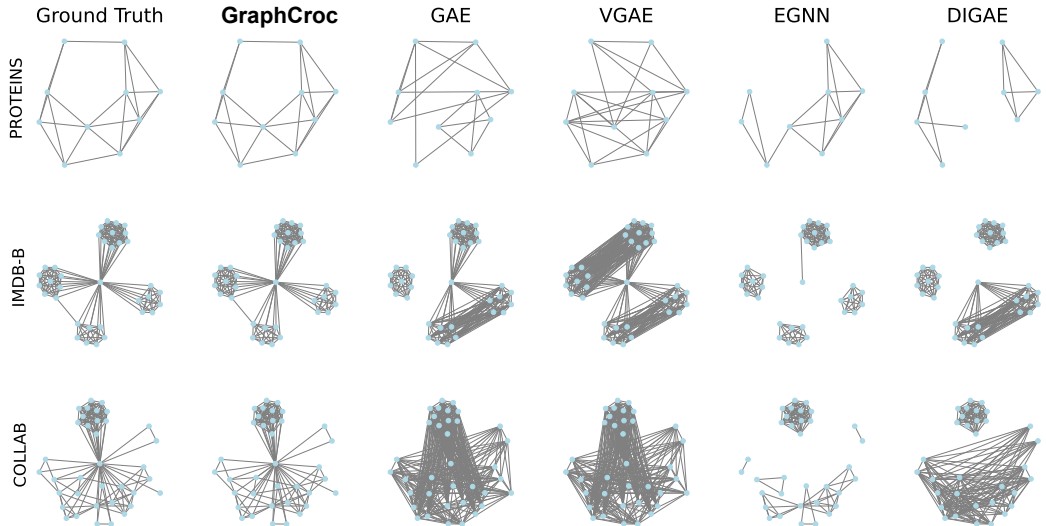

Figure 5: The reconstruction visualization. Other reconstructions are provided in Appendix G.5.

graphs, in the IMDB-B task. Our GraphCroc model significantly outperforms other GAE methods, while the self-correlation variant also delivers superior performance. Interestingly, while the L2-norm achieves a high AUC score, it is feeble to well reconstruct the entire graphs; this indicates there are some connection patterns in graph inherently hard to be captured by L2-norm representation. Other results of WL test are provided in Appendix G.3, which demonstrates similar observations as above.

In Figure 5, we select a representative graph from each of the PROTEINS, IMDB-B, and COLLAB tasks to visually compare the structural reconstruction from GAE models. It is evident that GraphCroc performs well in accurately recovering graph edges. GAE methods with inner-product and vanilla GNN architectures, such as GAE, VGAE, and DiGAE, tend to overpredict edge connections. Meanwhile, EGNN performs adequately within node clusters, but struggles with inter-cluster connections. This echos our discussion about the partial representation deficiency in L2-norm.

## 4.3 GraphCroc on Other GAE Strategies

To make cross-correlation more pronounced in our evaluation, the above experiments implement only the basic inner-product representation, i.e., sigmoid($PQ^T$). In addition, previous studies have extensively explored data augmentation and training enhancements to optimize GAE models. Specifically, we examine the performance of GAE with cross-correlation applied to different training strategies in Figure 6, using the PROTEINS dataset as a case study.

We evaluate three other prevalent decoding enhancements to compare their performance with the standard inner-product decoding (baseline) under the cross-correlation method. The variational decoding [19] generates node embeddings from a Gaussian distribution, with mean/std determined by the decoder outputs. Although a similar final AUC was achieved, it falls

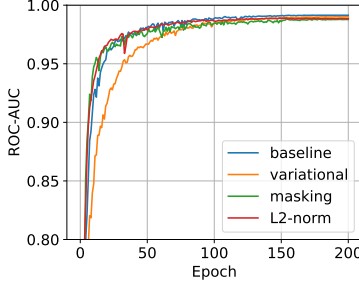

Figure 6: The AUC score of testing graphs in PROTEINS task, with different decoding methods.

short of the baseline on the PROTEINS task at early convergence. For the other two strategies, edge masking [35] and L2-norm representation [26], they facilitate faster convergence during the initial training stages. However, we find that the enhancement of these strategies is highly dependent on graph tasks. Our further analysis on other graph tasks (Appendix G.4) demonstrates that the L2-norm and masking could converge to worse structural reconstructions than our baseline training. Therefore, we still advocate our training and representing methods for GAE models on various graph tasks.

Table 2: Evaluation on graph classification tasks. We compare our model with other GNN methods, such as unsupervised learning (Infograph [34]), contrastive learning (GraphCL [48] and InfoGCL [44]), and generative learning (GraphMAE [15], S2GAE [35] and StructMAE [25]). The encoder of our model is extracted from GraphCroc, and is cascaded with a randomly-initialized 3-layer classifier. We train our models by only fine-tuning for 10 epochs or training for 100 epochs. The best results are in **bold**, and the second bests are underlined. Results are averaged on 5 runs.

| | Infograph | GraphCL | InfoGCL | GraphMAE | S2GAE | StructMAE | **ours** (10-epoch) | **ours** (100-epoch) |
|---|---|---|---|---|---|---|---|---|
| PROTEINS | 74.44 | 74.39 | – | 75.30 | 76.37 | 75.97 | $73.99^{\pm1.32}$ | $\mathbf{79.09}^{\pm1.63}$ |
| IMDB-B | 73.03 | 71.14 | 75.10 | 75.52 | 75.76 | 75.52 | $76.69^{\pm1.02}$ | $\mathbf{78.75}^{\pm1.35}$ |
| COLLAB | 70.65 | 71.36 | 80.00 | 80.32 | 81.02 | 80.53 | $81.70^{\pm0.54}$ | $\mathbf{82.40}^{\pm0.20}$ |

## 4.4 GraphCroc on Graph Classification Tasks

One common application of autoencoder models is leveraging their potent encoders for downstream tasks. We evaluate our GraphCroc model by employing its encoder in graph classification tasks, as summarized in Table 2. Notably, generative approaches like GraphMAE, S2GAE, StructMAE, and our GAE model tend to surpass traditional unsupervised and contrastive learning methods. Although contrastive learning incorporates negative sampling, its effectiveness is limited in multi-graph tasks. This finding corroborates the observations in Tab.4 of [44], which indicate that while negative sampling substantially boosts performance in single-graph tasks (e.g., node classification), it has little impact on graph classification tasks. In contrast, GAE models deliver robust graph representations, particularly for small-to-moderate-sized graphs, enhancing their utility in graph classification. Furthermore, our GraphCroc model significantly outperforms self-correlation methods (GraphMAE and S2GAE) in representing graph structures, demonstrated in Table 1, enabling the encoder to effectively capture the structural information of input graphs. Consequently, classifiers leveraging our encoder can achieve high performance with finetuning over only several epochs. Continued optimization of our classification models promises to further elevate their performance in graph classification tasks, surpassing other GAE-based models.

## 4.5 GAE Attack Surface Analysis

Research in vision tasks demonstrates that manipulating the latent space with perturbations enables AE to produce adversarial examples with stealthiness and semanticity [6, 16, 32, 40, 43]. Given AE's success in vision domain, we raise the concern — *whether a GAE can be utilized to generate adversarial graph structures by modifying the latent vectors?* Current graph adversarial attacks directly modify the input of GNNs, highly inefficient due to the discreteness of graph structures [7, 37]. By directly conducting attacks in the latent space, GAE could be a potentially efficient attack surface.

In Table 3, we demonstrate the effect of small perturbations on the latent space using random noise injection, PGD [27], and C&W adversarial noise injection [7] on graph classification tasks. We limit all attacks on the latent space with a maximum query number of $400$ and report the classification accuracy of perturbed graphs and the number of edge changes. Note that we focus solely on the structure modification without changes on the node features.

Table 3: Accuracy and modified edges for adversarial attack, leveraging latent perturbation on GAE. A lower accuracy indicates that the latent perturbation can better pass to the reconstructed graph. A higher percentage of modified edges indicates a larger attack cost.

| | Clean | Random | | PGD [27] | | C&W [4] | |
|---|---|---|---|---|---|---|---|
| | Acc. | Acc. | $\Delta$edge | Acc. | $\Delta$edge | Acc. | $\Delta$edge |
| IMDB-M | 55.3 | 53.5 | 4.79% | 45.7 | 24.5% | 39.7 | 17.4% |
| PROTEINS | 82.5 | 77.1 | 5.01% | 57.4 | 5.63% | 41.7 | 23.7% |
| COLLAB | 81.3 | 70.0 | 5.90% | 28.8 | 35.9% | 27.3 | 8.29% |

Our observations indicate that conducting adversarial attacks on the latent space of the graph autoencoder effectively reduces model accuracy, although it could yield significant edge changes. Compared to adversarial attacks on graph structures using discrete optimization methods, our latent attacks demonstrate comparable performance in terms of accuracy and can be performed in batches with high efficiency. Nevertheless, due to the discrete nature of graph structures, the distortion in edge changes is hard to be always controlled at a low level. Our evaluation of GraphCroc's latent space reveals a potential vulnerability, indicating that adversarial attacks on a graph autoencoder's latent space can provide efficient structural adversarial attacks. More detail on the adversarial attack with GAE is discussed in Appendix F.3.

## 5  Conclusion

Graph autoencoders (GAEs) are increasingly effective in representing graph structures. In our research, we identify significant limitations in the self-correlation approach employed in the decoding processes of prevalent GAE models. Self-correlation inadequately represents certain graph structures and requires optimization within a constrained space. To address these deficiencies, we advocate cross-correlation as the decoding kernel. We propose a novel GAE model, GraphCroc, which incorporates cross-correlation decoding and is built upon a U-Net architecture, enhancing the flexibility in GNN design. Our evaluations demonstrate that GraphCroc outperforms existing GAE methods in terms of graph structural reconstruction and downstream tasks. In addition, we propose the concern that well-performed GAEs could be a surface for adversarial attacks.

## Acknowledgments and Disclosure of Funding

We thank Shaolei Ren from UC Riverside for the valuable discussions that helped shape this work. This work is supported in part by the U.S. National Science Foundation under Grants OAC-2319962, CNS-2239672, CNS-2153690, CNS-2326597, CNS-2247892, and SaTC-1929300.

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

## A   Related Work

Graph representation has been explored through various methods. Auto-regressive models [24, 49, 14] generate graph structures by sequentially querying the connectivity between node pairs, which can be computationally expensive for large graphs, e.g., $n^2$ queries required for the adjacency matrix. Similarly, diffusion-based graph models [21] construct graph structures through multiple steps, such as degree matrix reconstruction [5]. These methods primarily focus on graph generation, creating rational graph structures from random noisy node islands.

In contrast, Graph Autoencoder (GAE) methods represent graph structures as node embeddings, designed to reconstruct the graph either sequentially [47, 22] or globally [33, 19, 15, 30, 20]. The very beginning graph structure representation is proposed in [19] with self-correlation (applied with Eq. 1 and 2) and further expresses the node embedding with a variational approach. Later, GAE has been widely explored with sequential and global generating methods. For the sequential GAE models, GraphRNN [47] proposes an autoregressive model, which generates graphs by summarizing a representative set of graphs and decomposes the graph generation into a sequence of node and edge formations. Similarly, [22] targets molecule generation and proposes to regard the graph structure as a parse tree from a context-free grammar, so that the VGAE can apply encoding and decoding to these parse trees. However, the sequential graph strategies usually are time-consuming or requiring expensive processing.

On the other hand, the global methods, which directly encode the graph in latent space and decode to the entire graph structure, have better scalability on larger and complicated graph structures and can be time efficient. GraphVAE [33] follows the VGAE idea and proposes an autoencoder model that can generate the graph structure, node features, and edge attributes all at once. EGNN [30, 26] decodes the node embedding to graph structures by applying the L2-norm between embeddings. GraphMAE [15] applies the masking strategy and targets to reconstruct the node features of various scales of graphs, where its GAE architecture also follows the classical GAE model. DiGAE [20] lies in the structural reconstruction on directed graphs, and firstly proposes to use the cross-correlation to express the node connection from two embedding spaces. Although these methods are effective recover node connections on a single graph, and even some of them tried the reconstruction of whole graph on graph tasks, there is no explicit evaluation to demonstrate how the global GAE model perform when it generate graph structure once and on moderate to large graph tasks. Besides, previous work follows the self-correlation strategy, which has been proven less effective than cross-correlation on graph tasks, in our work.

## B   Dimension Requirement to Well Represent Graph Structure

As the node embedding dimension $d'$ is underexplored before, it is mostly regarded as a hyperparameter to set up in advance. On the other hand, $d'$ highly effects the encoder representing ability, which is based on the graph scale. There is necessity to discuss the dimension requirement of node embeddings for a certain graph scale.

**Remark.** We share a toy example to demonstrate how the $d'$ design has an impact on the encoder ability. Assume an extreme case $d' = 1$, each node is represented by a scalar. The node embeddings $(z_i, z_j, z_k) \in \mathbb{R}^3$ can never represent a connection set $(A_{i,j}, A_{i,k}, A_{j,k}) = (0, 0, 0)$. Because if $\exists (z_i, z_j, z_k) \in \mathbb{R}^3$ such that $\mathcal{I}(\tilde{A}_{i,j}, \tilde{A}_{i,k}) = (0, 0)$, i.e., $z_i z_j < 0$ and $z_i z_k < 0$, then we must have $\text{sign}(z_j z_k) = \text{sign}(z_j z_i^2 z_k) = -1$. This will always yield to $\mathcal{I}(\tilde{A}_{j,k}) = 0 \neq A_{j,k}$. The similar result can be directly observed when $d' = 2$ and there are four nodes, then $\nexists (z_i, z_j, z_k, z_l) \in \mathbb{R}^4$ which can represent connection set $(A_{i,j}, A_{i,k}, A_{i,l}, A_{j,k}, A_{j,l}, A_{k,l}) = (0, 0, 0, 0, 0, 0)$.

**Lemma B.1.** *For self-correlation method in the decoder, to make the connection constraints always solvable on the $n$-node scenario, i.e., requiring $z_i^T z_j > 0$ or $z_i^T z_j < 0$ for each node pair, the node embedding dimension $d'$ need to satisfy $d' \geq (n-1)$ at least.*

*Proof.* We first prove that for $n$ nodes, there is always existing a connection case, such as no connection on all node pairs, that $\nexists \{z\} \in \mathbb{R}^{n \times d'}$ can represent when $d' < (n-1)$. We consider the case that $A_{i,i} = 1$ and $A_{i,j} = 0$ for all $i, j \in [1, n]$, such as $z_i^T z_j < 0$. When $d' < (n-1)$, e.g., $d' = (n-2)$, there will be at most $(n-2)$ linearly independent node vectors. Assume the first $(n-2)$ vectors $z_1$ to $z_{n-2}$ are linearly independent, then we will always find a linear combination

such that $\sum_{i=1}^{n-1} \alpha_i z_i = \mathbf{0}$, where vector $\boldsymbol{\alpha} = \{\alpha_1, ..., \alpha_{n-1}\} \neq \mathbf{0}$. Here, we let the elements of $\boldsymbol{\alpha}$ be grouped as positives, negatives, and zeros, as $\boldsymbol{\alpha}^+, \boldsymbol{\alpha}^-, \boldsymbol{\alpha}^0$. Thus, we have

$$\sum_{i=1}^{n-1} \alpha_i z_i = \mathbf{0} \Rightarrow \sum_{a \in \boldsymbol{\alpha}^+} a_i z_i = \sum_{b \in \boldsymbol{\alpha}^-} -b_j z_j = \boldsymbol{w}$$

**a)** If both $\boldsymbol{\alpha}^+, \boldsymbol{\alpha}^-$ are not empty, we do inner product on these two terms:

$$0 < \boldsymbol{w}^T \boldsymbol{w} = \left( \sum_{a \in \boldsymbol{\alpha}^+} a_i z_i \right)^T \left( \sum_{b \in \boldsymbol{\alpha}^-} -b_j z_j \right)$$
$$= \sum -a_i b_j \cdot z_i^T z_j \leq 0$$

This causes conflict on the inequality. **b)** If one of $\boldsymbol{\alpha}^+$ and $\boldsymbol{\alpha}^-$ is empty, e.g., $\boldsymbol{\alpha}^- = \emptyset$, we do inner product between the positive part and $z_n$:

$$0 = \mathbf{0}^T z_n = \sum_{a \in \boldsymbol{\alpha}^+} a_i z_i^T z_n < 0$$

which also cause inequality conflict. Thus, the assumption on $d' < (n-2)$ does not hold.

Then, we prove that when $d' = (n-1)$, there always exists $\{z\} \in \mathbb{R}^{n \times d'}$ to represent all the connections through the decoder. We use **inductive method** to prove it. It is straightforward that when $n = 2$, $d' = 1$ can satisfy the representation on any graph $A \in \{0, 1\}^{2 \times 2}$. Assuming $d' = n-1$ is a feasible configuration for an arbitrary $n$-node graph, we need to prove that $d' = n$ is also sufficient for $(n+1)$-node graph: We denote a satifiable node embedding from the $n$-node graph as $Z = \{z_i\} \in \mathbb{R}^{n, n-1}$. By extending it to $d' = n$ for one more node coming, we specify the node embedding of the first $n$ nodes as $z_i' = [z_i, 0]$ for $i = [1, n-1]$ and $z_n' = [z_n, 1]$; this specification can still satisfy arbitrary inequality constraints between node pairs in the first $n$ node. For the $(n+1)$-th node, we need to find $z_{n+1}' \in \mathbb{R}^n$ such that $z_{n+1}'^T z_i'$ satisfy arbitrary inequalities. Through the inductive method, it is also straightforward to prove that there exists $Z$ with rank $(n-1)$, thus our extension to one extra dimension will make $\text{rank}(\{z_i'\}) = n$ for $i = [1, n]$. Therefore, we can always find a non-zero vector $\boldsymbol{\alpha}$ such that $z_{n+1}' = \sum_{i=1}^{n} \alpha_i z_i'$. For each constraint $z_{n+1}'^T z_i'$ being positive or negative (in total $n$ constraints), we can reduce them to system of equations where the constants are scalars satisfying the constraints:

$$\left. \begin{array}{c} \alpha_1 z_1'^T z_1 + \quad ... \quad +\alpha_n z_n'^T z_1 = c_1 \\ ... \\ \alpha_1 z_1'^T z_n + \quad ... \quad +\alpha_n z_n'^T z_n = c_n \end{array} \right\} n \text{ equations}$$

denoted as $M\boldsymbol{\alpha} = C$. Here, the vector $\boldsymbol{\alpha}$ is solvable as long as $\text{rank}(M) = \text{rank}(M|C)$, which can be tuned by specifying $C$.

Although this theoretical analysis indicates that the node embedding dimension should be large enough to ensure the graph structure reconstruction, we observed in experiments that the embedding dimension can usually be smaller (e.g., $d' \approx n/2$ because the hard-to-solve structures are not common in graph tasks. Nevertheless, it provides a preterior node embedding dimension suggestion, and our evaluation widely adopts this lemma and takes $d' \approx n$ in all experiments.

## C  Specific Graph Structure Representation

In Sec. 2.2 and 2.3, we explore the limitations of self-correlation and the effectiveness of cross-correlation in expressing specific graph structures. Given that previous GAE research often evaluates undirected asymmetric graph structures, the evaluation on special graph structures is usually overlooked. Hereby we evaluate how our method GraphCroc and other GAE models perform on specific graph structures as aforementioned.

**Island (without self-loop) and symmetric graph structure.** We generate 4 topologically symmetric graphs devoid of self-loops. Thus, the task is to have the evaluated GAE learn to reconstruct these graph structures and assess their performance. The visualization of their reconstruction is presented

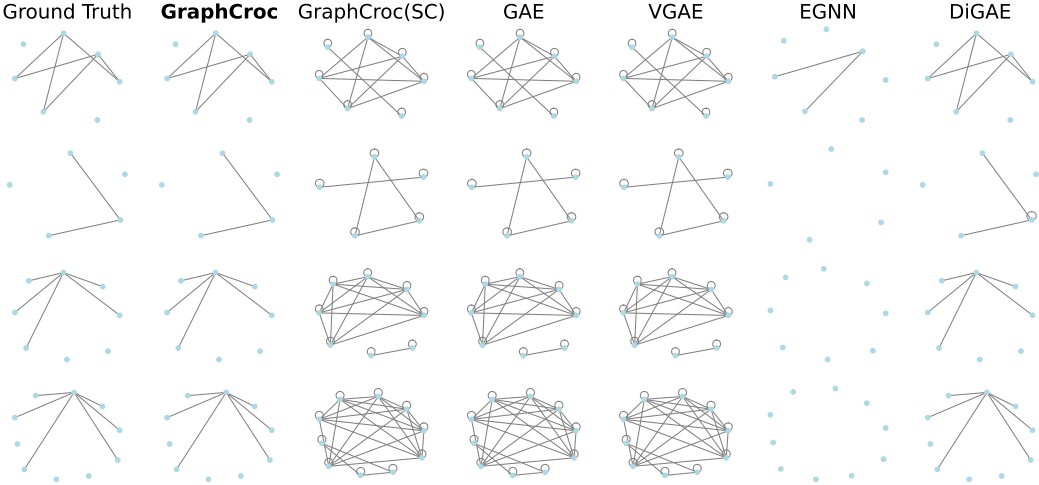

Figure 7: The graph reconstruction visualization of different models on graphs with symmetric structure and non-self-looped nodes.

in Fig. 7. The visualization clearly demonstrates that our GraphCroc model effectively reconstructs these specialized graph structures. For DiGAE which is also based on cross-correlation, it can also well reconstruct the special graph structures, further supporting our discussion in Sec. 2.3. In contrast, other self-correlation-based models tend to incorrectly predict connections between symmetric nodes and islands, and incorrectly introduce self-loops on nodes. Note that for EGNN, it does not predict positive edges between nodes, which seems not to follow our analysis in Sec. 2.2 with Euclidean encoding $\text{sigmoid}(C(1 - \|z_i - z_j\|^2))$. This is because EGNN slightly improves this encoding to $\text{sigmoid}(w\|z_i - z_j\|^2 + b)$, where $w$ and $b$ are learnable. Since no-self-loop nodes require $\text{sigmoid}(w\|z_i - z_i\|^2 + b) = \text{sigmoid}(b) < 0.5$, $b$ is forced to be negative, inducing negative prediction on symmetric edges that have $z_i = z_j$ under symmetric structures. Therefore, EGNN still cannot handle well the graph reconstruction on the special graph structures.

**Directed graph structure.** We conduct an evaluation using datasets of directed graphs. We compare GraphCroc with DiGAE, as only cross-correlation-based methods are capable of expressing directional relationships between nodes. To construct the dataset, we sample subgraphs from the directed Cora_ML [28] and CiteSeer [10] datasets. Specifically, we randomly select 1,000 subgraphs. Of these, 800 subgraphs were used for training and 200 for testing. The results are detailed in Table 4, where $\bar{N}$ represents the average number of nodes per graph:

Table 4: The AUC score of reconstructing the adjacency matrix in directed graph tasks.

|  | Cora_ML($\bar{N} = 41$) | Cora_ML($\bar{N} = 77$) | CiteSeer($\bar{N} = 16$) |
|---|---|---|---|
| DiGAE | 0.6879 | 0.8296 | 0.9083 |
| GraphCroc | 0.9946 | 0.9996 | 0.9999 |

Our GraphCroc model can well reconstruct the directed graph structure with almost perfect prediction, which significantly outperforms the DiGAE model. This advantage comes from the expressive model architecture of our proposed U-Net-like model.

# D  Node Embedding Divergence in GraphCroc Decoder

The difference between two latent embeddings (denoted as $P$ and $Q$) is fundamental to cross-correlation as opposed to self-correlation in which $P = Q$; therefore, it is necessary to make them not converge to each other. One method of explicitly controlling this divergence is by incorporating regularization terms into the loss function, such as cosine similarity ($\cos(P, Q)$).

Our decoder architecture inherently encourages differentiation between $P$ and $Q$ since they are derived from two separate branches of the decoder. This structure can allow $P$ and $Q$ to diverge adaptively in response to the specific needs of the graph tasks. If a graph cannot be well

represented by self-correlation, our two-branch structure will encourage sufficient divergence on $P$ and $Q$ to suit structural reconstruction. To evaluate the differentiation between them, we compute their cosine similarity and present a histogram of these values for each graph task in Figure 8. Across all tasks, the cosine similarity between the node embeddings under cross-correlation is generally low, typically below 0.6. This shows that our two-branch decoder effectively maintains the independence of the node embeddings, which are adaptively optimized for various graph tasks. Furthermore, this adaptive optimization underscores the superiority of cross-correlation in real-world applications, as evidenced by GraphCroc's superior performance in graph structural reconstruction compared to other methods (Table 1).

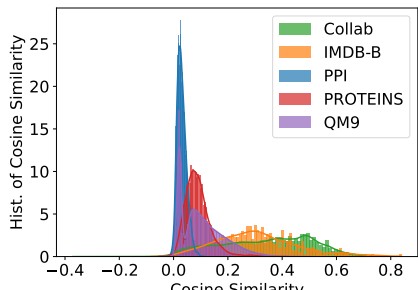

Figure 8: The histogram of cosine similarity between node embeddings $P$ and $Q$ under cross-correlation, applying GraphCroc on graph tasks.

## E    Loss Balancing Derivation

We adopt binary cross-entropy (BCE) loss to evaluate reconstruction between prediction $\tilde{A} = \mathrm{sigmoid}(PQ^T)$ and ground truth $A$. Our loss balancing is based on an i.i.d. assumption between $\mathcal{L}_0$ and $\mathcal{L}_1$, where the loss definition follows $\mathcal{L}(i,j) = BCE(\tilde{A}_{i,j}, A_{i,j})$ on the node pair $(i,j)$. For a certain graph $G$, we assume there are $c_0$ zero elements and $c_1$ one elements in $A$, where $c_0 \gg c_1$. To balance the loss between zeros and ones, we apply scaling factors on each element loss: $\mathcal{L}(\tilde{A}, A) = \alpha_0 \sum_{i=1}^{c_0} \mathcal{L}_i^0 + \alpha_1 \sum_{j=1}^{c_1} \mathcal{L}_j^1$. The scaling has two targets: to keep the loss magnitude and to balance the zero/one influence. Thus, we construct the following linear equations:

$$\begin{cases} \alpha_0 c_0 \mathcal{L}^0 + \alpha_1 c_1 \mathcal{L}^1 = c_0 \mathcal{L}^0 + c_1 \mathcal{L}^1 \\ \alpha_0 c_0 \mathcal{L}^0 = \alpha_1 c_1 \mathcal{L}^1 \end{cases} \Rightarrow \alpha_0 = \frac{c_0 + c_1}{2c_0}, \quad \alpha_1 = \frac{c_0 + c_1}{2c_1}$$

The scaling factors $\alpha_0$ and $\alpha_1$ are derived.

## F    Supplementary Experimental Setup

### F.1    Dataset

We provide the graph detail of graph tasks selected in our evaluation, in Table 5. For IMDB-B and COLLAB without node features, we take the one-hot encoding of degree as the node features.

Table 5: The dataset configuration of selected graph tasks.

|          | # graphs | # nodes (avg) | # edges (avg) | # features | # classes/tasks |
|----------|----------|---------------|---------------|------------|-----------------|
| PROTEINS | 1,113    | 39.1          | 145.6         | 3          | 2               |
| IMDB-B   | 1,000    | 19.8          | 193.1         | 0          | 2               |
| COLLAB   | 5,000    | 74.5          | 4914.4        | 0          | 3               |
| PPI      | 22       | 2245.3        | 61,318.4      | 50         | 121             |
| QM9      | 130,831  | 18            | 37.3          | 11         | 19              |

### F.2    Other Experimental Setup Information

We provide a three-layer GraphCroc to demonstrate the detailed data flow and the model structure, in Figure 9. Besides, we list the detailed configuration of GraphCroc model and corresponding training setup for all graph tasks in Table 6. The reconstruction results in Table 1 are not provided in average on multiple runs, because the reproduction on several experiments is too heavy loaded. For example, due to the large graph size, the default setting (vector dimension of 128 and layer number of 4) in EGNN when reproducing the PPI task will cause the out-of-memory issue on the 40GB A100 GPU. While reducing the dimension to 16 and the layer number to 3 allows model and data to fit just into GPU (38.40GB/40GB), this significantly simplified model fails to adequately learn the structure of the PPI graphs and performs poorly compared to other GAE methods. In addition, given that graph reconstruction must be conducted by graph and QM9 task has massive graphs, even one model training on QM9 will take over 2 GPU days.

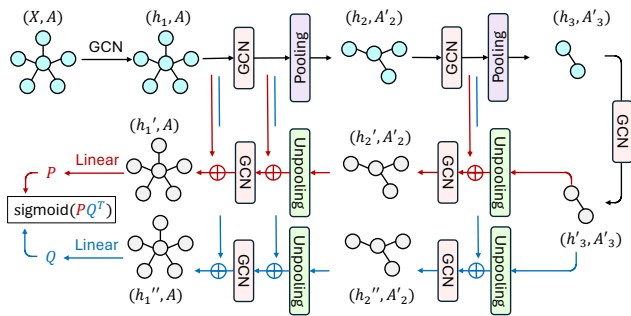

Figure 9: The archtecture example of our GraphCroc model, with 3-layer encoder/decoder.

Table 6: The architecture and training configuration of GraphCroc on selected graph tasks.

| | input dim. | embedding dim. | # layers | pooling rate | training config. (opt., lr, epochs) |
|---|---|---|---|---|---|
| PROTEINS | 3 | 128 | 7 | [-, 0.9, 0.8, 0.7, 0.6, 0.5, -] | (AdamW, 1e-3, 200) |
| IMDB-B | 135 | 128 | 7 | [-, 0.9, 0.8, 0.7, 0.6, 0.5, -] | (AdamW, 1e-3, 200) |
| COLLAB | 400 | 128 | 8 | [-, 0.9, 0.8, 0.7, 0.6, 0.5, 0.4, -] | (AdamW, 1e-3, 200) |
| PPI | 50 | 1024 | 11 | [-, 0.5, 0.5, 0.5, 0.5, 0.5, 0.5, 0.5, 0.5, -] | (AdamW, 1e-3, 200) |
| QM9 | 11 | 32 | 6 | [-, 0.9, 0.8, 0.7, 0.6, -] | (AdamW, 1e-3, 100) |

### F.3  Adversarial Attack on GraphCroc

In Section 4.5, we evaluate the GAE performance as an adversarial attack surface. Specifically, given a pretrained encoder $\Phi(Z|G)$ which encodes the graph into a latent space and a downstream classifier $f(Z)$ for the graph classification task, we aim to generate a perturbed latent representation $Z'$ and leverage a reconstructor $\Theta$ to rebuild the graph structure $G' = (X, A')$. The goal of the adversarial structure is to cause the encoder and downstream classifier to misclassify the graph, i.e., $f(\Phi(G')) \neq y$, where $y$ is the original label, and the only difference between $G$ and $G'$ is the adjacency matrix $A$. We assess two gradient-based adversarial attacks on latent space.

**Projected Gradient Descent (PGD) [27]**: PGD iteratively perturbs the input to maximize the classification loss of $f(Z)$:

$$\delta_{t+1} = \mathbf{Proj}_{||\delta||_1 \leq \epsilon}(\delta_t + \alpha \cdot \text{sign}(\nabla_{\delta_t}\mathcal{L}(f(Z), y)))$$

**Carlini & Wagner (C&W) [4]**: C&W finds adversarial latent vectors by solving an optimization problem:

$$\delta^* = \arg\min_\delta ||\delta||_1 + c \cdot (\max f(Z)_y - \max_{i \neq y}(f(Z)_i, -k))$$

Here, $f(Z)_y$ denotes the logits output of the classifier for the true class $y$, and $k$ is a confidence parameter that controls the confidence of the misclassification. This optimization can be solved with gradient descent. Hence, the final adversarial graph structure will be $G' = (X, \Theta(Z + \delta^*))$. To enhance the performance of adversarial perturbation, we fine-tune the reconstructor $\Theta$ during the adversarial attack. Specifically, to ensure the effectiveness of the reconstructed adversarial example, we optimize $\Theta$ by minimizing the distance between the perturbed latent representation and the latent space of the reconstructed graph structure:

$$\Theta^* = \arg\min_\Theta ||Z + \delta^* - \Phi(X, \Theta(Z + \delta^*))||$$

## G  Supplementary Experiment Results

### G.1  Structural Reconstruction of Cross-Correlation on Other Architectures

In addition to the GCN kernel used in our GraphCroc model, we extend our analysis to include other widely used graph architectures such as GraphSAGE [13], GAT [36], and GIN [45]. To incorporate these architectures into the cross-correlation framework, we replace the GCN module with corresponding operations while preserving the overarching structure, which includes the encoder,

the dual-branch decoder, and the skip connections between the encoder and decoder. Furthermore, we explore how GraphCroc performs without skipping connections. The overall architecture and training configurations remain consistent with those outlined in Table 6 of our paper. The results, presented in Table 7, follow the format of Table 1 in our paper, providing a clear comparison across different architectures.

Overall, all architectures employing cross-correlation effectively reconstruct graph structures, underscoring the significance of cross-correlation as a core contribution of our work. Given that training each model requires several hours, particularly for large datasets such as PPI and QM9, we do not fine-tune the hyperparameters much during model training. The results presented here may represent a lower bound of these archi-

Table 7: The AUC score of reconstructing the adjacency matrix in graph tasks, under different architectures in the cross-correlation framework.

| Dataset | GraphSAGE | GAT | GIN | GraphCroc* | GraphCroc |
|---|---|---|---|---|---|
| PROTEINS | 0.9898 | 0.9629 | 0.9927 | 0.9934 | 0.9958 |
| IMDB-B | 0.9984 | 0.9687 | 0.9980 | 0.9975 | 0.9992 |
| Collab | 0.9985 | 0.9627 | 0.9954 | 0.9976 | 0.9989 |
| PPI | 0.9774 | 0.9236 | 0.9467 | 0.9447 | 0.9831 |
| QM9 | 0.9972 | 0.9978 | 0.9974 | 0.9966 | 0.9987 |

* without skip connection

tectures' potential performance. Therefore, we refrain from ranking these cross-correlation-based architectures due to their closely matched performance, and we adopt a conservative stance in our comparisons. Nevertheless, it is evident that most of these architectures (except GAT) generally surpass the performance of self-correlation models shown in Table 1 of our paper, highlighting the efficacy of cross-correlation in graph structural reconstruction.

## G.2 Node Trajectory during Training

In Figure 10, we demonstrate the converging trajectory of first tow nodes of eight other graphs during training, as an extension of Figure 2. It aligns with the analysis in main paper that cross-correlation can provide a much smoother than self-correlation during the GAE training.

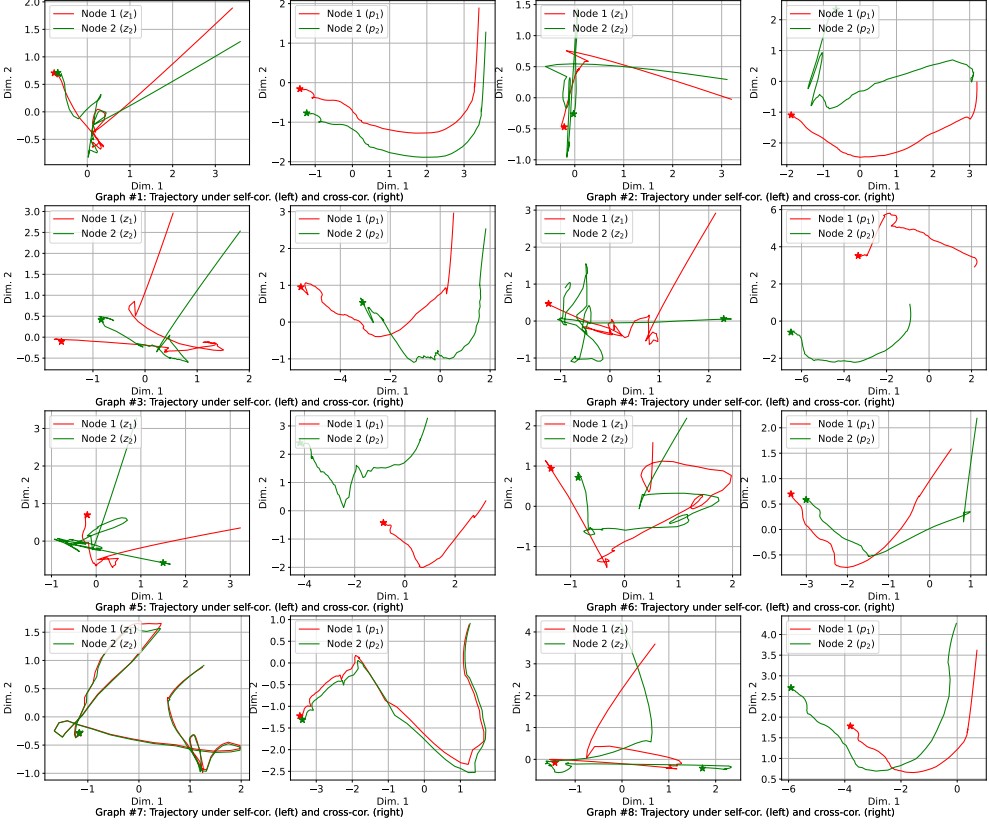

Figure 10: The trajectory of the first two nodes of eight graph samples during training.

## G.3 WL-Test Results

In Figure 11, we provide the WL-test results on graph tasks, PROTEINS, COLLAB, PPI, and QM9. Our GraphCroc can still outperform other GAE models in completely reconstructing the graphs in most tasks. Notably, all the methods cannot achieve the isomorphism on PPI reconstruction. This is because PPI only has two test graphs, while they have number of nodes 3324 and 2300, which are too large to be completely reconstructed.

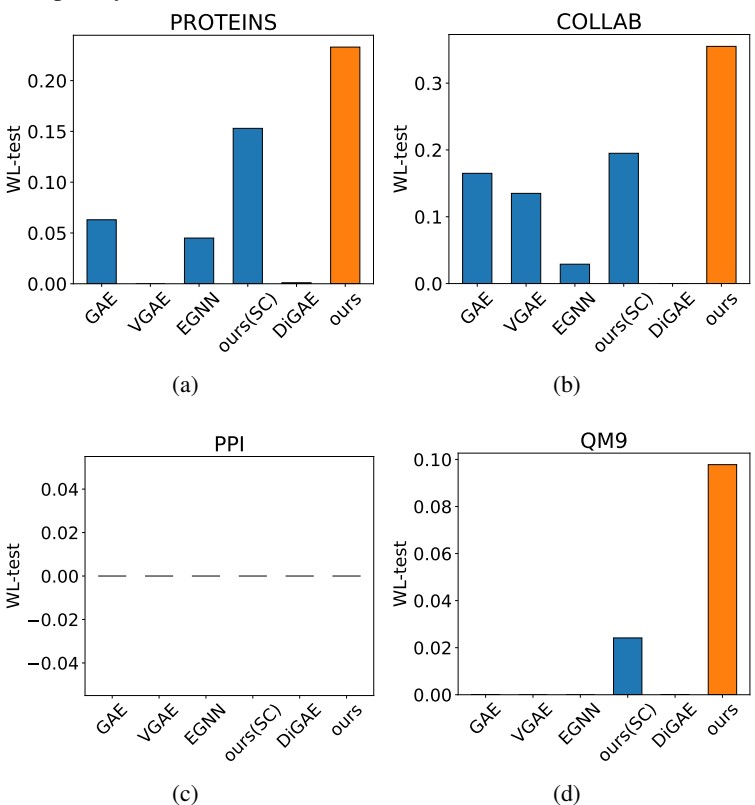

Figure 11: The WL-test results on PROTEINS, COLLAB, PPI, and QM9. "ours" refers to GraphCroc.

## G.4 GraphCroc on Other GAE Strategies

In Figure 12, we demonstrate more evaluations for GraphCroc on GAE strategies. The AUC score is further tested on IMDB-B, COLLAB, and PPI tasks. We find that the GAE training integrated with other enhancements, i.e., variational, edge masking, and L2-norm, performs distinctively on different graph tasks. They could underperform our baseline training strategy on certain tasks.

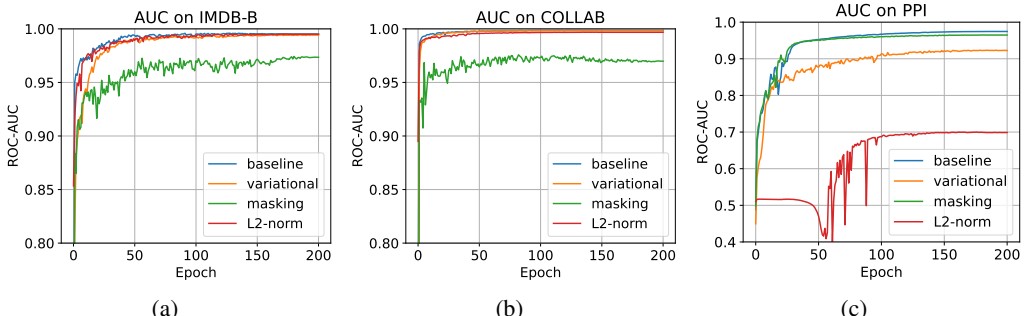

Figure 12: The supplementary AUC scores on other graph tasks, with different decoding methods.

## G.5 Reconstruction Visualization

We visualize more graph structure reconstruction results on graph tasks, in Figure 13, 14, 15, and16.

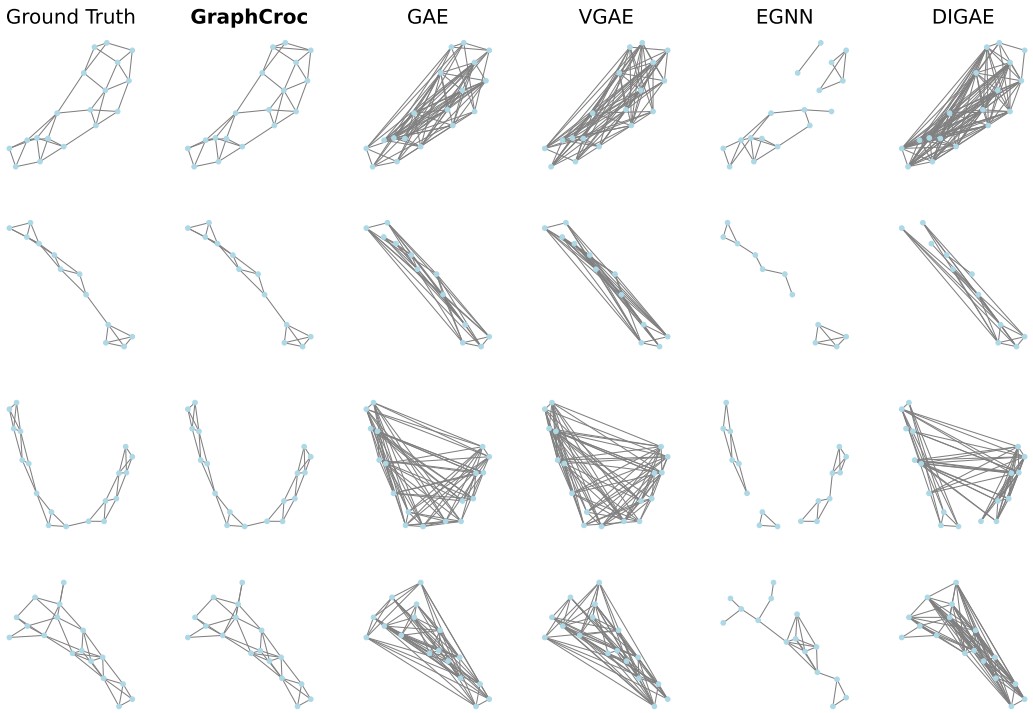

Figure 13: The structure reconstruction visualization on PROTEINS task.

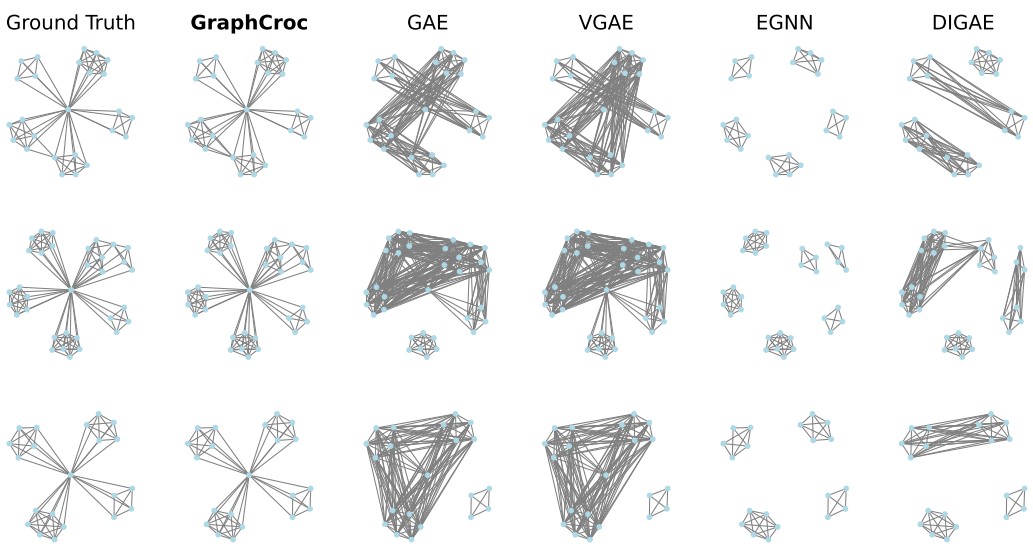

Figure 14: The structure reconstruction visualization on IMDB-B task.

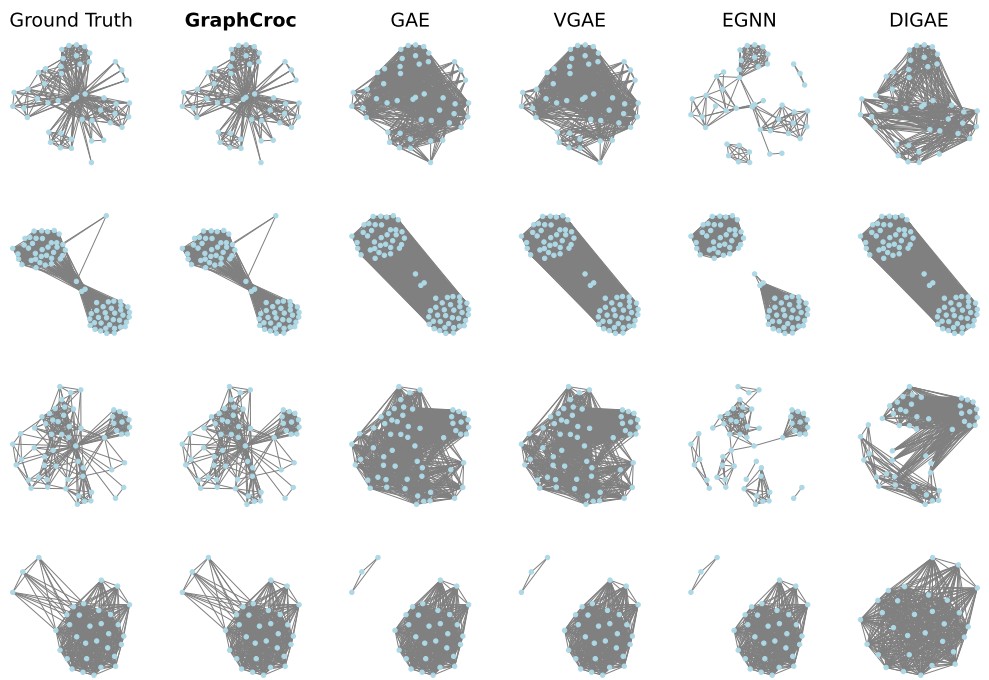

Figure 15: The structure reconstruction visualization on COLLAB task.

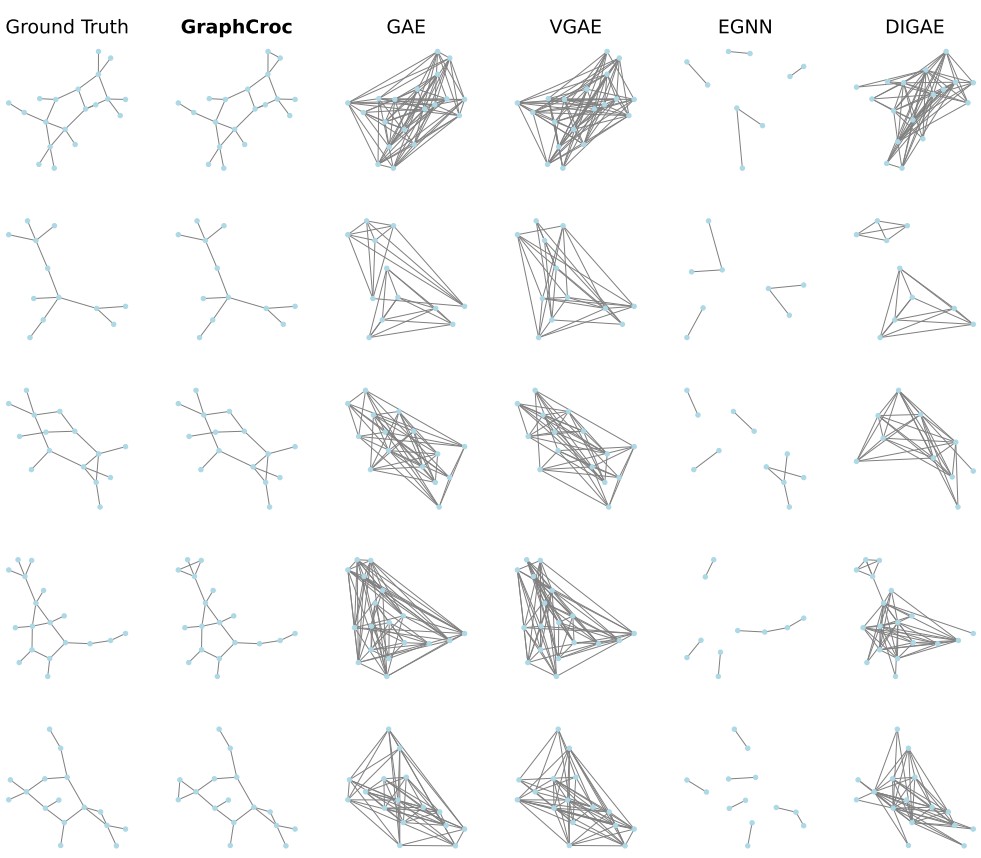

Figure 16: The structure reconstruction visualization on QM9 task.

