# OpenReview forum: "GraphCroc: Cross-Correlation Autoencoder for Graph Structural Reconstruction"
_NeurIPS.cc/2024/Conference — NeurIPS 2024 poster_

### Official Review · Reviewer_QrYC · 2024-07-11

**Soundness:** 3
**Presentation:** 3
**Contribution:** 3
**Rating:** 5
**Confidence:** 4

**Summary:**

This paper proposes a cross-correlation autoencoder for graph structural reconstruction. The authors first analyze the problems of existing self-correlation encoder. Then, a cross-correlation autoencoder is designed. Experimental results show the effectiveness of the cross-correlation autoencoder.

**Strengths:**

1. The motivation is clear and the cross-correlation autoencoder is reasonable.
2. The paper is well-written and easy to follow.
3. The experiments are comprehensive.

**Weaknesses:**

1. The authors mention that the current self-correlation methods can not address specific (sub)graph structures. But this paper only presents an overall experimental performance. It is unclear how the proposed cross-correlation autoencoder performs given a specific graph structure.

2. It is not clear whether the graph dataset used in the paper is a directed or undirected graph. Since the cross-correlation autoencoder can represent the directed graph effectively, it is suggested to consider the directed graph dataset.

3. More different architectures of the encoder and decoder should be employed to further verify the effectiveness of the cross-correlation mechanism.

**Questions:**

see Weakness.

---

> ### Author Rebuttal · Authors · 2024-08-06
>
> > Evaluate the proposed cross-correlation autoencoder given specific graph structures, e.g., islands and symmetric structures.
>
> In Sec.2.2 and 2.3, we explore the limitations of self-correlation and the capabilities of cross-correlation in accurately representing specific graph structures, such as non self-loop, symmetric structures, and directed edges. In the evaluation, we follow previous GAE research to apply our proposed GraphCroc on common real-world graph tasks (Table 1), which are undirected asymmetric graph structures. Additionally, we present a comparison of how various models reconstruct specific graph structures in `Fig. 1` of our rebuttal PDF file. Specifically, we randomly generate 4 graphs where each graph is *topologically symmetric* and contains *no self-loop*. These graphs are then used to evaluate the performance of different GAE models. The visualizations clearly illustrate that our GraphCroc model proficiently reconstructs these graph structures. For DiGAE which is also based on cross-correlation, it can well reconstruct the special graph structures, further proving our discussion in Sec.2.2 and 2.3.
>
> In contrast, other models often erroneously predict connections between symmetric nodes and introduce unwanted self-loops, highlighting the superior representation ability of cross-correlation in handling these specialized scenarios. Note that for EGNN, it does not predict positive edges between nodes, which seems not to follow our analysis in Sec.2.2 with Euclidean encoding (sigmoid$(C(1-||z_i-z_j||^2))$, $C>0$) [6]. This is because EGNN slightly improves this encoding to sigmoid$(w||z_i-z_j||^2+b)$ where $w$ and $b$ are learnable. Since no-self-loop nodes require sigmoid$(w||z_i-z_i||^2+b)=$sigmoid$(b)<0.5$, $b$ is forced to be negative, inducing negative prediction on symmetric edges which have $z_i=z_j$ under symmetric sturctures; alternatively, we can also regard it as the naive Euclidean encoding but with $C<0$. Therefore, EGNN still cannot handle well the graph reconstruction on the special graph structures.
>
> [6] Graph normalizing flows, NeurIPS'19.
>
> > Evaluation on direction graphs.
>
> Most graph reconstruction research is evaluated on undirected tasks, so for a fair and comprehensive comparison, we also utilize undirected graphs. Additionally, structural reconstruction on directed graphs has gained focus recently in the DiGAE work (AAAI'22). Hereby, we further investigate GraphCroc's performance on directed graph datasets. We compare GraphCroc with DiGAE, given that only cross-correlation-based methods can effectively capture directional relationships between nodes. To construct the dataset, we sample subgraphs from the directed Cora and Citeseer datasets. We randomly selected 1,000 subgraphs, using 800 for training and 200 for testing. The results are shown below ($\bar{N}$ represents the average number of nodes per graph):
>
> || cora ($\bar{N}=41$) | cora ($\bar{N}=77$) | cite ($\bar{N}=16$) |
> |-|:-:|:-:|:-:|
> | GraphCroc| 0.9946  | 0.9996  | 0.9999  |
> | DiGAE    | 0.6870  | 0.8296  | 0.9083  |
>
> Our GraphCroc can well reconstruct the directed graph structure with almost perfect prediction, which significantly outperforms the DiGAE model. This advantage comes from the expressive ability of our proposed U-Net-like architecture.
> > More different architectures of the encoder and decoder should be employed to further verify the effectiveness of the cross-correlation mechanism.
>
> Thank you for your suggestion of further evaluation on cross-correlation across different GNN architectures. In addition to the GCN kernel used in our GraphCroc model, we extend our analysis to include other widely used graph architectures such as GraphSAGE [7], GAT [8], and GIN [9]. To incorporate these architectures into the cross-correlation framework, we replace the GCN module with corresponding operations while preserving the overarching structure, which includes the encoder, the dual-branch decoder, and the skip connections between the encoder and decoder. Furthermore, we explore how GraphCroc performs without skipping connections. The overall architecture and training configurations remain consistent with those outlined in Table 5 of our paper, except for the QM9 dataset, where we limit the training to 20 epochs due to its large size and the limited time frame for our rebuttal. The results, presented below, follow the format of Table 1 in our paper, providing a clear comparison across different architectures:
>
> | | GraphSAGE| GAT| GIN|GraphCroc(w/o skip connection)| GraphCroc|
> |-|:-:|:-:|:-:|:-:|:-:|
> | PROTEINS | 0.9898   | 0.9629  | 0.9927  |  0.9934 | 0.9958 |
> | IMDB-B   | 0.9984   | 0.9687  | 0.9980  |  0.9975 | 0.9992 |
> | Collab   | 0.9985   | 0.9627  | 0.9954  |  0.9976 | 0.9989 |
> | PPI      | 0.9774   | 0.9236  | 0.9467  |  0.9447 | 0.9831 |
> | QM9      | 0.9972   | 0.9978  | 0.9974  |  0.9966 | 0.9987 |
>
> Overall, all architectures employing cross-correlation effectively reconstruct graph structures, underscoring the significance of cross-correlation as a core contribution of our work. Given that training each model requires several hours, particularly for large datasets such as PPI and QM9, we did not fine-tune the hyperparameters much during model training. The results presented here may represent a *lower bound* of these architectures' potential performance. Therefore, we refrain from ranking these cross-correlation-based architectures due to their closely matched performance, and we adopt a conservative stance in our comparisons. Nevertheless, it is evident that most of these architectures (except GAT) generally surpass the performance of self-correlation models shown in Table 1 of our paper, highlighting the efficacy of cross-correlation in graph structural reconstruction.
>
> [7] Inductive representation learning on large graphs, NeurIPS'17;
>
> [8] Graph attention networks, ICLR'18;
>
> [9] How powerful are graph neural networks?, ArXiv'18.

---

> > ### Author Response · Authors · 2024-08-13
> >
> > Dear reviewer QrYC,
> >
> > As the rebuttal discussion is about to close, we would like to confirm whether our rebuttal has adequately addressed your concerns. If there are any questions you would like to discuss, please let us know.

---

### Official Review · Reviewer_Xxf1 · 2024-07-12

**Soundness:** 3
**Presentation:** 3
**Contribution:** 3
**Rating:** 6
**Confidence:** 4

**Summary:**

This paper proposed a method to address the limitations of existing graph autoencoder (GAE) models that primarily rely on self-correlation for graph structure representation. They claim existing GAE often fail to accurately represent complex structures like islands, symmetrical structures, and directional edges, particularly in smaller or multiple graph contexts. The proposed model, GraphCroc, introduces a cross-correlation mechanism that aims at enhancing the representational capabilities of GAEs. It employs a mirrored encoding-decoding process to ensure robust structural reconstruction and introduces a loss-balancing strategy to tackle representation bias during optimization.

**Strengths:**

1. The idea to introduce two latent space for reconstructing the graph structure is "simple and intuitive".

2. The writing is clear and easy to follow.

3. The experimental results are sound.

**Weaknesses:**

1. This paper lacks discussion on related works. There already exists some works trying to solve the graph autoencoder structure recovering issues. For example, including position encoding [1] or adding extra node labels [2]. How the proposed method is compared with these methods, from the perspective of effectiveness and efficiency?

[1] You, Jiaxuan, Rex Ying, and Jure Leskovec. "Position-aware graph neural networks." International conference on machine learning. PMLR, 2019.

[2] M. Zhang, P. Li, Y. Xia, K. Wang, and L. Jin, Labeling Trick: A Theory of Using Graph Neural Networks for Multi-Node Representation Learning, Advances in Neural Information Processing Systems (NeurIPS-21), 2021.

2. As the proposed method generate two latent embeddings, I wonder if there exists some techniques to control them to be different with each other? Otherwise I am concerned that whether the two embeddings could converge to each others.

**Questions:**

see above weakness

---

> ### Author Rebuttal · Authors · 2024-08-06
>
> > This paper lacks discussion on related works. There already exists some works trying to solve the graph autoencoder structure recovering issues. For example, including position encoding or adding extra node labels. How the proposed method is compared with these methods, from the perspective of effectiveness and efficiency?
>
> Thank you for your suggestion. The first referenced paper incorporates positional information into node embeddings, whereas the second paper explores the effectiveness of the labeling trick by adding labels to nodes of interest. Before delving into comparisons, it is necessary to highlight a shared point across these studies and ours: **all approaches try to introduce asymmetry into node embeddings, to effectively represent node connections in symmetric/isomorphic graph structures**. In our work, this is achieved through cross-correlation and the application of two-set node embeddings. This shared point is also applied in the DiGAE approach [5], although it is not explicitly discussed in their work, given their focus on representing asymmetric directed graphs. This commonality makes these methods all theoretically effective for representing special graph structures, such as symmetric nodes; yet this effectiveness is limited to their application scenarios.
>
> Our GraphCroc model is designed to represent the entire graph all at once, directly outputting the complete predicted adjacency matrix rather than individual predicted edges. This makes GraphCroc particularly suited for per-graph downstream tasks such as graph classification. In contrast, the labeling trick is designed for link prediction, where only specific node pairs or subsets are labeled, creating asymmetry between these nodes and others. Labeling must be conducted either pair-by-pair or subset-by-subset to maintain this asymmetry, which is a critical aspect of the labeling trick's approach. If all nodes were labeled to reconstruct all edges, distinctions would be needed between specific node labels to break down symmetry effectively. However, GraphCroc offers an easier approach, reconstructing the graph structure in one go without the need for repeated generation of node embeddings for different node pairs. This efficiency makes GraphCroc more effective for whole-graph reconstruction, while the labeling trick remains better suited for tasks focusing on link prediction between selected node pairs.
>
> Position Encoding (PGNN) is also evaluated in link prediction tasks. However, PGNN has the potential to be applied to structural reconstruction across the entire graph, provided the selected anchor set facilitates asymmetric message aggregation. Regarding efficiency, PGNN's process must regenerate the anchor set of the newly structured graph after each pooling layer, due to the dynamic nature of graph structures in GNN tasks, which can lead to notable inefficiencies. In contrast, GraphCroc does not need preprocessing on the graph structure, but dynamically adjusts asymmetry through the parameters of its decoder during training; thus, GraphCroc offers greater efficiency and adaptability. Furthermore, the asymmetry in PGNN is predetermined by the anchor set selection and remains static throughout training, making it dependent on initial anchor choices. This contrasts with GraphCroc's more flexible and adaptive approach to handling graph structure representations.
>
> In conclusion, a significant application difference is that GraphCroc encodes the entire graph in a latent embedding suitable for downstream tasks, allowing the inclusion of graph pooling layers. However, both position encoding and the labeling trick primarily focus on node-level embeddings, preserving the graph's original structure, with downstream tasks limited to node classification and link prediction.
>
> [5] Directed graph auto-encoders, AAAI'22.
>
>
> >  As the proposed method generate two latent embeddings, I wonder if there exists some techniques to control them to be different with each other? Otherwise I am concerned that whether the two embeddings could converge to each others.
>
> The difference between two latent embeddings (denoted as $P, Q$) is fundamental to cross-correlation as opposed to self-correlation in which $P=Q$; therefore, it is necessary to make them not converge to each other. One method of explicitly controlling this divergence is by incorporating regularization terms into the loss function, such as cosine similarity ($cos(P,Q)$).
>
> However, our decoder architecture inherently encourages differentiation between $P$ and $Q$ since they are derived from two separate branches of the decoder. This structure can allow $P$ and $Q$ to diverge adaptively in response to the specific needs of the graph tasks. If a graph cannot be well represented by self-correlatin, our two-branch structure will encourage sufficient divergence on $P$ and $Q$ to suit structural reconstruction. To evaluate the differentiation between $P$ and $Q$, we compute their cosine similarity and present a histogram of these values for each graph task in `Fig. 2` of our rebuttal PDF file. Across all tasks, the cosine similarity between the node embeddings under cross-correlation is generally low, typically below 0.6. This shows that our two-branch decoder effectively maintains the independence of the node embeddings, which are adaptively optimized for various graph tasks. Furthermore, this adaptive optimization underscores the superiority of cross-correlation in real-world applications, as evidenced by GraphCroc's superior performance in graph structural reconstruction compared to other methods (Table 1 of our paper).

---

> > ### Author Response · Authors · 2024-08-13
> >
> > Dear reviewer Xxf1,
> >
> > As the rebuttal discussion is about to close, we would like to confirm whether our rebuttal has adequately addressed your concerns. If there are any questions you would like to discuss, please let us know.

---

> > > ### Comment · Reviewer_Xxf1 · 2024-08-13
> > > **Thank you for your response**
> > >
> > > Thank the authors for their explanations, which have mostly addressed my concerns on the related works and divergence of two generated latent embeddings. I encourage the authors to include the discussion of related works into their camera ready/future revisions. Here I keep my positive scores.

---

> > > > ### Author Response · Authors · 2024-08-13
> > > > **Thanks for your recognition!**
> > > >
> > > > Thank you for recognizing our work! We will include the discussion of related work and the divergence of latent embeddings in our camera-ready/revisions.

---

### Official Review · Reviewer_Msno · 2024-07-25

**Soundness:** 3
**Presentation:** 2
**Contribution:** 3
**Rating:** 5
**Confidence:** 4

**Summary:**

This paper theoretically analyzes the limitations of existing graph autoencoders (GAE) in representing special graph features such as islands, symmetrical structures, and directional edges. To address this, the paper proposes a new GAE method, GraphCroc, which employs a cross-correlation mechanism that significantly enhances the representational capabilities of GAEs.

**Strengths:**

1. The paper clearly shows the limitations of existing GAEs through theoretical analysis.

2. The experimental results demonstrate the advantages of the proposed method in structural reconstruction and graph classification tasks.

3. The paper is easy to follow.

**Weaknesses:**

1. In Table 1, the improvements of GraphCroc are evident only on two datasets.

2. While the proposed cross-correlation method performs better than the general self-correlation method on island, symmetric structures, and directed graphs, it would be beneficial to include more results in reconstruction visualization, particularly regarding island or directed edge reconstruction.

3. Some related works [1] need to be discussed.

[1] Liu, Chuang, et al. "Where to Mask: Structure-Guided Masking for Graph Masked Autoencoders." arXiv preprint arXiv:2404.15806 (2024).

**Questions:**

1. How about the performance of the proposed method on directed graphs?

---

> ### Author Rebuttal · Authors · 2024-08-06
>
> > In Table 1, the improvements of GraphCroc are evident only on two datasets.
>
> AUC is widely used to evaluate graph structural reconstruction tasks in GAE, due to its unbiased performance on positive and negative edges. Thus, we adopt this metric to assess the adjacency matrix reconstruction in our work. GraphCroc exhibits only marginal improvements on Collab and QM9 graph tasks compared to EGNN [1]. This can be attributed to the good representation capabilities of EGNN on certain graph tasks. Specifically, EGNN achieves AUC over 0.99 on these tasks, nearing perfect prediction (AUC=1), thus leaving minimal room for improvement by GraphCroc. However, Table 1 also illustrates the significant advancements of GraphCroc over EGNN on other graph tasks like PROTEINS and IMDB-B. Moreover, EGNN was not successfully applied to the PPI task due to the out-of-memory issue (on a 40GB A100 GPU), whereas GraphCroc operates within a mere 3GB of memory, as we measured. This demonstrates not only the generality of GraphCroc in handling various real-world tasks but also its efficiency in implementation. Additionally, the core of our paper focuses on the cross-correlation mechanism, which inherently offers greater expressiveness than self-correlation models like EGNN.
>
> *Note:* Regarding the performance, we have some further comments. Observing the self-correlation models, both EGNN and GraphCroc(SC) can achieve good AUC scores, only slightly lower than our proposed cross-correlation-based GraphCroc, yet GAE and VGAE have poor performance. This is because GAE/VGAE [2] was proposed early and has a simple architecture, while EGNN and GraphCroc(SC) (U-Net-like architecture [3]) are more complicated and have more powerful representation ability on graph structures, outperforming normal GAE/VGAE methods. On the other hand, *common graph tasks are usually asymmetric*, and *self-loops are overlooked* in previous works, which is why even the self-correlation-based EGNN and GraphCroc(SC) can still perform well on these graphs reconstruction. Nevertheless, cross-correlation can still boost the graph structural reconstruction on these graphs, as evidenced by the best performance of our GraphCroc on all graph tasks.
>
> [1] E (n) equivariant graph neural networks, ICML'21;
>
> [2] Variational graph auto-encoders, NeurIPS'16;
>
> [3] Graph u-nets, ICML'19.
>
> > More results in reconstruction visualization for specific graph structures, such as island, symmetric structures, and directed graphs.
>
> In Sec.2.2 and 2.3, we explore the limitations of self-correlation and the effectiveness of cross-correlation in expressing specific graph structures. Given that previous GAE research often evaluates undirected asymmetric graph structures, we similarly provide the evaluation of graph structural reconstruction for common real-world graph tasks (Table 1). Additionally, we agree that it would be highly beneficial to include clear visualizations demonstrating how various encoding methods succeed or fail in reconstructing specific graph structures.
>
> **island (without self-loop) and symmetric graph structure:** We generate 4 topologically symmetric graphs devoid of self-loops. We task the evaluated models with learning to reconstruct these graph structures and assess their performance. The visualization of their reconstruction is presented in `Fig. 1` of our rebuttal PDF file. The visualization clearly demonstrates that our GraphCroc model effectively reconstructs these specialized graph structures. For DiGAE which is also based on cross-correlation, it can also well reconstruct the special graph structures, further proving our discussion in Sec.2.2 and 2.3. In contrast, other self-correlation-based models tend to incorrectly predict connections between symmetric nodes and islands, and incorrectly introduce self-loops on nodes. Note that for EGNN, it does not predict positive edges between nodes, which seems not to follow our analysis in Sec.2.2 with Euclidean encoding (sigmoid$(C(1-||z_i-z_j||^2))$) [4]. This is because EGNN slightly improves this encoding to sigmoid$(w||z_i-z_j||^2+b)$ where $w$ and $b$ are learnable. Since no-self-loop nodes require sigmoid$(w||z_i-z_i||^2+b)=$sigmoid$(b)<0.5$, $b$ is forced to be negative, inducing negative prediction on symmetric edges that have $z_i=z_j$ under symmetric structures. Therefore, EGNN still cannot handle well the graph reconstruction on the special graph structures.
>
> **directed graph structure:** We conduct an evaluation using datasets of directed graphs. We compare GraphCroc with DiGAE, as only cross-correlation-based methods are capable of expressing directional relationships between nodes. To construct the dataset, we sample subgraphs from the directed Cora and CiteSeer datasets. Specifically, we randomly select 1,000 subgraphs. Of these, 800 subgraphs were used for training and 200 for testing. The results are detailed below, where $\bar{N}$ represents the average number of nodes per graph:
>
> |          | Cora_ML ($\bar{N}=41$) | Cora_ML ($\bar{N}=77$) | CiteSeer ($\bar{N}=16$) |
> |---|:-:|:-:|:-:|
> | GraphCroc| 0.9946  | 0.9996  | 0.9999  |
> | DiGAE    | 0.6870  | 0.8296  | 0.9083  |
>
> Our GraphCroc can well reconstruct the directed graph structure with almost perfect prediction, which significantly outperforms the DiGAE model. This advantage comes from the expressive model architecture of our proposed U-Net-like model.
>
> [4] Graph normalizing flows, NeurIPS'19.
>
> > Related work discussion: "Where to Mask: Structure-Guided Masking for Graph Masked Autoencoders." IJCAI'24.
>
> Thanks for your suggestion about the related work. This work addresses node importance in graph construction and proposes a structure-based masking strategy. This strategy guides the masking on GAE with more rationality, and is well evaluated based on the standard GraphMAE. We will give a discussion of this work and include its performance on graph classification tasks in Table 2 of our paper.

---

> > ### Author Response · Authors · 2024-08-13
> >
> > Dear reviewer Msno,
> >
> > As the rebuttal discussion is about to close, we would like to confirm whether our rebuttal has adequately addressed your concerns. If there are any questions you would like to discuss, please let us know.

---

### Author Rebuttal · Authors · 2024-08-06

We appreciate the time and effort the reviewers have spent in providing valuable feedback! We are grateful for the reviewers' recognition of our clear writing, reasonable motivation, and sound experiments. Graph structural reconstruction is a pivotal application for graph autoencoders (GAEs), and we hope that our research offers a novel perspective on graph representation and proves beneficial to the community.

In response to your comments, we address all identified weaknesses and questions. We welcome further feedback and are eager to engage in discussions regarding the paper's content. Additionally, following suggestions for enhanced visualizations, we include new figures in our rebuttal and attach them in the PDF file below. References to these figures are highlighted, e.g., `Fig. 1`. All responses to reviewers' comments will be reflected in our paper's final version.

---

> ### Author Response · Authors · 2024-08-11
>
> Dear reviewers,
>
> Again we appreciate your time and effort in reviewing our paper. We hope that we have adequately addressed your concerns and would value your confirmation in this regard. We also look forward to your feedback on our rebuttal and are open to any further discussion you may have.
>
> Best regards,
>
> Authors

---

### Decision · Program_Chairs · 2024-09-25

**Decision:**

Accept (poster)

**Comment:**

The reviewers unanimously recommend acceptance of the paper with varying degrees of strength. I agree with their assessment and am happy to accept this paper for publication at the NeurIPS conference.

The reviews and the rebuttal have given rise to some interesting points and results that I encourage the authors to include in the camera-ready version. In particular, the experiments on directed graphs and different encoder architectures further the empirical evidence in favour of the proposed approach.